# CSV: Content Service Offloading System with Vehicular Caching

**DOI:** 10.3390/s22207967

**Published:** 2022-10-19

**Authors:** Yeunwoong Kyung, Taewon Song

**Affiliations:** 1Division of Information & Communication Engineering, Kongju National University, Cheonan 31080, Korea; 2Department of Internet of Things, Soonchunhyang University, Asan 31538, Korea

**Keywords:** content delivery network, vehicular caching, service continuity, service migration

## Abstract

Vehicular caching (VC) has been considered as a promising technology to provide low end-to-end service latency and reduce the load of networks. However, it is difficult for VC to provide service continuity because of its opportunistic availability according to mobility. To mitigate this problem, we introduce a CSV: Content service offloading System with VC which can opportunistically distribute the load of the content server and support the service continuity. In CSV, the content service can be migrated between fog node (FN) and VC while maintaining the ongoing service without service disruption, which can opportunistically distribute the load of the content server and support the service continuity during migration. To assess the performance of CSV, we develop an analytical model for VC offloading efficiency. Extensive simulation results demonstrate that CSV can reduce the load of the content server compared to the conventional system.

## 1. Introduction

Recently, video traffic has explosively grown as numerous Internet of Things (IoT) devices consume massive amounts of video content such as higher definition 360-degree videos and augmented/virtual reality services [1]. To provide low end-to-end service latency and reduce the load of networks, contents are shifted from cloud to fog nodes (FNs) co-located with the access network device such as a cellular base station (BS) [2]. However, even though the FNs are promising to deliver the contents to users, the scalability issue can also occur because of limited resources and increasing number of content service users including IoT devices [3]. For instance, if many IoT devices within a similar location request content simultaneously, FN can be overloaded. To mitigate this challenge, content delivery via vehicular caching (VC) is introduced. VC can store the contents, and thus users can receive contents from distributed VC, which can reduce the load of FN. VC can provide content to users only when they are in the vicinity of users and have content that the users request [4,5]. This means that users can opportunistically receive the contents from VC when both conditions are satisfied.

For efficient utilization of VC, numerous works have been conducted to determine the content placement and resource allocation [5,6], provide the collaborative content dissemination [7,8], and consider social and economic attributes [4,9]. In these works, the content delivery within the connection time between vehicles is usually utilized. This means that these works did not consider the service continuity between the connection and disconnection time according to the mobility of VC. On one hand, when VC provides the contents to users, service disruption can unwillingly occur because of its mobility during the service. In other words, the ongoing session between the user and VC can be terminated when VC leaves, which can result in the quality of service (QoS) degradation. On the other hand, when static FN delivers the contents to users, service offloading might be required when VC becomes available to distribute the load of FN. Therefore, how VC can be exploited with service continuity as much as possible considering its mobility should be analyzed to efficiently distribute the load of FN.

Meanwhile, there have been works to optimize the service provisioning based on the users’ mobility. As an initial concept, follow-me-cloud has been introduced to allow cloud services to be migrated to the optimal data center according to the users’ mobility without service disruption [10]. In addition to cloud services, this concept has been extended to FN architecture [11,12]. However, these works have focused on the mobility of users. This means that the mobility of the content source such as VC is not taken into account. Although our previous work introduced a policy at each decision epoch to select VC or FN to provide contents to users [13], it cannot make the best use of VC and does not consider the service continuity for the content service offloading in the continuous time domain.

To offload more content services to VC with service continuity, we introduce the content service offloading system with VC, named CSV. In CSV, the content service can be migrated between FN and VC while maintaining the ongoing service without service disruption. That is, when VC moves close to the user and has the same content that the user requests, the ongoing content service is offloaded from FN to VC. Then, the content service can be provided through VC without affecting the user, which also makes the best use of VC. As the 3GPP standards have made progress in supporting vehicular communications, vehicle-to-everything (V2X) services, and vertical enabler architectures [14,15], CSV can be one of the network scenarios in the future mobile networks (i.e., 6G). The key contribution of this paper is twofold: (1) to the best of our knowledge, this is the first analytical study that considers the service continuity for the content service offloading with VC. Specifically, we develop the analytical models for the VC offloading efficiency considering the service continuity; and (2) based on the simulation works, we demonstrate how much content can be delivered with VC, which can be interpreted as the load distribution effect.

The remainder of this paper is organized as follows. The system description is introduced in Section 2. In Section 3, the performance analysis is conducted. Then, the simulation results and conclusion are provided in Section 4 and Section 5, respectively.

## 2. System Description

Figure 1 shows the system model of this paper. It is assumed that FN connected to BS is always available whereas VC is only available when it moves close enough to users. In addition, the contents that users request can exist or not in VC according to the popularity of contents and the limited caching capacity of VC. Although there can be a caching capacity constraint for FN, we assume that all the contents for users are cached in FN to focus on the load distribution effect by VC. Considering the central cloud node to request the contents from FN will be one of our future works. Note that users can recognize the connection and disconnection with VC by using periodic signaling messages [16,17,18]. Moreover, for the service migration, this paper utilizes session migration between FN and VC which transfers the session information such as socket and state information to provide light and fast service continuity [12,19]. Since the session migration is not for the full content synchronization, it can be performed between nodes that have the same content currently being served.

As an example in Figure 1, users 1 and 2 want to download the specific content. In the case of user 1, the session starts at $\tau_{s1}$. At the time, since VC is not available, the content service is provided directly from FN. After $\tau_{1}$, to utilize VC which becomes available for user 1, the service migration is performed from FN to VC, which results in the period of no change for the downloaded content size, $t_{m}$. After the migration, the content service is continuously provided to user 1 from VC until the end of the session $\tau_{e1}$ without service disruption. On the other hand, the session of user 2 starts at $\tau_{s2}$ with VC and content service can be provided from VC because of its proximity. The session should be maintained after VC leaves at $\tau_{4}$. In this case, after the service migration from VC to FN is performed (i.e., $t_{m}$), user 2 can continuously use the content service from FN until the end of session $\tau_{e2}$. In this way, CSV can opportunistically utilize the VC which can distribute the load of FN while maintaining the ongoing content service without affecting users.

For the tactical analysis, we assume that the contact time between VC and the user (e.g., from $\tau_{1}$ to $\tau_{2}$ and from $\tau_{3}$ to $\tau_{4}$ in Figure 1) denoted by $t_{v}$ follows an exponential distribution with mean 1/$\mu_{v}$ as it is widely utilized for VC residence time [20]. In addition, it is assumed that the period when VC is not available (i.e., non-contact time) represented by $t_{f}$ follows an exponential distribution with mean 1/$\mu_{f}$ as it is usually assumed for the period in a FN (BS)-only area [21,22]. From the start, the session continues during the session duration represented by $t_{d}$ (i.e., $t_{d1}$ and $t_{d2}$ in Figure 1) which follows an exponential distribution with mean 1/$\lambda_{d}$ [23]. During the session duration, the content service can be provided to the user. The elapsed time from the session start to the migration is represented by $t_{rf}$ (i.e., from the session start with FN and migration to VC) and $t_{rv}$ (i.e., from the session start with VC and migration to FN) as shown in Figure 1. According to the residual life theory [24], the PDF of $t_{rf}$ and $t_{rv}$ can be given by
(1)$$f_{rf}\left(t\right)=\mu_{f}\left(1-F_{f}\left(t\right)\right)$$
and
(2)$$f_{rv}\left(t\right)=\mu_{v}\left(1-F_{v}\left(t\right)\right).$$

The VC offloading time denoted by $t_{off}$ is defined as the elapsed session duration in the contact time between VC and the user (e.g., from $\tau_{1}$ to $\tau_{2}$ and from $\tau_{3}$ to $\tau_{4}$ in Figure 1). The VC offloading time can be exploited to provide the content service to users through VC, which consequently can reduce the load on FN. Based on the assumption that the user meets a VC once in session duration, $t_{off}$ is less or equal to $t_{v}$. In addition, we assume that FN and VC can deliver the contents to users with the data rate of BS (i.e., $d_{b}$) and that of VC (i.e., $d_{v}$), respectively.

Let *Q* denote the total number of contents. Since we assume that the popularity of the content follows a Zipf distribution [5,25], the request probability of content *q* is given by
(3)$$p_{q}=\frac{1}{{r_{q}}^{\epsilon }}/\left(\sum_{q=1}^{Q}\frac{1}{{r_{q}}^{\epsilon }}\right)$$
where ϵ is the exponent characterizing the distribution and $r_{q}$ is the popularity ranks of the content *q*. Moreover, due to the limited caching capacity of VC (i.e., $C_{v}$), the least-recently-used (LRU) is assumed as it is generally utilized in the content caching process for the replacement policy [26].

## 3. Performance Analysis

In this section, the analytical model of CSV is introduced in order to evaluate the performance of CSV compared to that of the non-migration system. In the non-migration system, service migration is not considered. That is, when the session starts with FN, it maintains with FN during the session duration even though VC becomes available. On the other hand, if the session starts with VC, it terminates when VC leaves and a new session re-starts with FN. Specifically, VC offloading efficiency, $E_{E}$, which is defined as the ratio of the expected amount of downloaded content from VC to the expected total amount of downloaded content during the session period, is derived as an important performance metric.

### 3.1. Effective VC Offloading Time

To obtain the offloading efficiency by VC of CSV, the expected VC offloading time, $E[t_{off}]$, can be derived considering five disjoint cases in Figure 2: (a) $\tau_{s}\in t_{f}$, $t_{d}$ < $t_{rf}$, (b) $\tau_{s}\in t_{f}$, $t_{rf}\leq t_{d}<t_{rf}+t_{v}$, (c) $\tau_{s}\in t_{f}$, $t_{rf}+t_{v}\leq t_{d}$, (d) $\tau_{s}\in t_{v}$, $t_{d}<t_{rv}$, and (e) $\tau_{s}\in t_{v}$, $t_{rv}\leq t_{d}$. Let $E_{Y,C}$ be the partial offloading time in the case Y∈ {a, b, c, d, e} for CSV. Consequently, $E_{Y,C}$ can be derived as follows.

#### 3.1.1. $E_{a,C}$

When $\tau_{s}\in t_{f}$, $t_{d}$ < $t_{rf}$, the content service is provided only by FN because the session starts and ends before VC is available. Therefore, $E_{a,C}$ is simply 0.

#### 3.1.2. $E_{b,C}$

When $\tau_{s}\in t_{f}$, $t_{rf}\leq t_{d}<t_{rf}+t_{v}$, the ongoing session is migrated to VC and completed during the VC contact time $t_{v}$. As a result, $t_{off}$ is given by $t_{d}-t_{rf}-t_{m}$. The joint PDF of the $t_{d}$, $t_{rf}$, and $t_{v}$ is obtained by $f_{d,rf,v}\left(t_{d},t_{rf},t_{v}\right)$. Since $t_{d}$, $t_{rf}$, and $t_{v}$ are independent, the joint PDF can be the product of each PDF (i.e., $f_{d}\left(t_{d}\right)f_{rf}\left(t_{rf}\right)f_{v}\left(t_{v}\right)$). Moreover, since $\int_{t_{v}=0}^{\infty }\int_{t_{rf}=0}^{\infty }f_{rf}\left(t_{rf}\right)f_{v}\left(t_{v}\right)dt_{rv}dt_{v}$ is commonly required for $E_{b,C},E_{c,C},E_{d,C}$, and $E_{e,C}$, we omit the common part for simplicity. Consequently, $E_{b,C}$ can be defined as
(4)$$E_{b,C}=\int_{t_{d}=t_{rf}+t_{m}}^{t_{rf}+t_{v}}\left(t_{d}-t_{rf}-t_{m}\right)f_{d}\left(t_{d}\right)dt_{d}.$$

Note that $E_{b,C}$ is valid when $t_{d}-t_{rf}\geq t_{m}$. In other words, if $t_{d}-t_{rf}<t_{m}$, $E_{b,C}$ is assumed to be 0 (i.e., $t_{off}$ is not effective) due to the long migration time, which can be considered as an offloading failure. The offloading failure will be described in Section 3.3.

#### 3.1.3. $E_{c,C}$

When $\tau_{s}\in t_{f}$, $t_{rf}+t_{v}\leq t_{d}$, the ongoing session is migrated to VC and completed after the VC contact time $t_{v}$. Consequently, $t_{off}$ is obtained by $t_{v}-t_{m}$. Therefore, $E_{c,C}$ can be given by
(5)$$E_{c,C}=\int_{t_{d}=t_{rf}+t_{v}}^{\infty }\left(t_{v}-t_{m}\right)f_{d}\left(t_{d}\right)dt_{d}.$$

Note that $E_{c,C}$ is valid when $t_{v}>t_{m}$. In other words, if $t_{v}\leq t_{m}$, $E_{c,C}$ is assumed to be 0 (i.e., $t_{off}$ is not effective), which can be considered as an offloading failure and also will be described in Section 3.3.

#### 3.1.4. $E_{d,C}$

If $\tau_{s}\in t_{v}$, $t_{d}<t_{rv}$, the session starts and ends during the VC contact time, which means that $t_{off}$ is same to $t_{d}$. As a result, $E_{d,C}$ can be obtained by
(6)$$E_{d,C}=\int_{t_{d}=0}^{t_{rv}}t_{d}f_{d}\left(t_{d}\right)dt_{d}.$$

#### 3.1.5. $E_{e,C}$

If $\tau_{s}\in t_{v}$, $t_{rv}\leq t_{d}$, $t_{off}$ is same with $t_{rv}$ because the session starts in the VC contact time and ends after the contact time. Therefore, $E_{e,C}$ can be defined
(7)$$E_{e,C}=\int_{t_{d}=t_{rv}}^{\infty }t_{rv}f_{d}\left(t_{d}\right)dt_{d}.$$

Because the cases from (a) to (e) are disjoint events, the expected VC offloading time of CSV, $E[t_{off}]$, can be obtained by $E_{a,C}+E_{b,C}+E_{c,C}+E_{d,C}+E_{e,C}$.

On the other hand, $E[t_{off}]$ in the non-migration system, can be the same with the sum of $E_{d,C}$ and $E_{e,C}$ because it does not consider the service migration when the VC becomes available. Specifically, when the session starts with FN (like cases from (a) to (c) in Figure 2), the session maintains with FN (i.e., no migration to VC) until it completes. Therefore, the offloading by VC in the non-migration system is only valid when the session starts during VC contact time.

To utilize the expected VC offloading time (i.e., $E[t_{off}]$) as derived above for both systems, it is required for VC to have the same content as that of the user’s ongoing session. Note that the content *q* can be requested by VC with the request probability $p_{q}$ and then be cached at VC. Therefore, the effective VC offloading time, $E_{T}$, can of CSV can be given by $p_{q}E\left[t_{off}\right]$.

### 3.2. VC Offloading Efficiency

The expected amounts of downloaded content during the effective VC offloading time (i.e., $E_{T}$) can be calculated as $E_{T}d_{v}$ for both CSV and non-migration systems. On the other hand, the expected total amounts of downloaded content during the session duration can be obtained by $E_{T}d_{v}+\left(E\left[t_{d}\right]-E_{T}\right)d_{b}$. Therefore, the VC offloading efficiency can be given by
(8)$$E_{E}=\frac{E_{T}d_{v}}{E_{T}d_{v}+\left(E\left[t_{d}\right]-E_{T}\right)d_{b}}.$$

Particularly, if $E_{E}=0$, it means that the content of the user’s ongoing session does not exist in VC, or the session starts and ends without VC offloading. On the other hand, if $E_{E}=1$, it indicates that the session maintains with VC through the entire session duration.

### 3.3. VC Offloading Failure

As described in Section 3.1, when the user tries to offload the service from FN to VC considering two cases (b) and (c) in Figure 2, it can be failed due to the long migration time. Let $F_{Y}$ be the probability of the VC offloading failure in the case Y∈{b,c{ for CSV. Consequently, $F_{Y}$ can be derived as follows.

#### 3.3.1. $F_{b}$

In case (b), the VC offloading failure can occur when $t_{d}-t_{rf}<t_{m}$. Consequently, $F_{b}$ can be obtained by
(9)$$F_{b}=\int_{t_{v}=0}^{\infty }\int_{t_{rf}=0}^{\infty }\int_{t_{d}=t_{rf}}^{t_{rf}+t_{m}}f_{d}\left(t_{d}\right)f_{rf}\left(t_{rf}\right)f_{v}\left(t_{v}\right)dt_{d}dt_{rf}dt_{v}.$$

#### 3.3.2. $F_{c}$

In case (c), if $t_{v}\leq t_{m}$, the VC offloading failure can occur. Therefore, $F_{c}$ can be defined as
(10)$$F_{c}=\int_{t_{v}=0}^{t_{m}}\int_{t_{rf}=0}^{\infty }\int_{t_{d}=t_{rf}+t_{v}}^{\infty }f_{d}\left(t_{d}\right)f_{rf}\left(t_{rf}\right)f_{v}\left(t_{v}\right)dt_{d}dt_{rf}dt_{v}.$$

Because the cases (b) and (c) are disjoint events, the total probability of the VC offloading failure for CSV, $P_{F}$, can be obtained by $F_{b}+F_{c}$.

## 4. Simulation Results

To verify the analytical results, event-driven simulations based on MATLAB R2018b are performed in Windows 10 with 16 GB RAM. Moreover, arrival times of events are drawn by generating 20,000 random numbers of $t_{f}$ and $t_{d}$ and the average values are utilized. As described in Section 3, the simulation results of CSV are compared to those of the non-migration scheme. The default values of $\mu_{f}$, $\lambda_{d}$, $d_{b}$, $d_{v}$, *Q*, $t_{m}$, $r_{q}$, and $C_{v}$ are 10, 10, 10 Mbps, 5 Mbps, 50, 0.01, 1, and 10 GB respectively [9,25] as shown in Table 1. In addition, the content size is within the range of [500 MB, 1000 MB] and the exponent of the Zipf distribution (i.e., ϵ) is 0.7 [25]. Figure 3 shows the effect of the average contact time between VC and the user $1/\mu_{v}$ which is normalized by the non-contact time $1/\mu_{f}$ for the VC offloading efficiency $E_{E}$ and effective VC offloading time $E_{T}$. In Figure 3a,b, as the VC contact time increases, VC offloading efficiency becomes higher because the expected VC offloading time can increase. From Figure 3a, it can be found that CSV can achieve higher offloading efficiency compared to the non-migration system. This is because CSV efficiently exploits the VC to offload the service when it becomes available thanks to the service migration. In addition, VC offloading efficiency at $1/\lambda_{d}=0.1$ is higher than that at $1/\lambda_{d}=0.2$. This is because the increasing amount of downloaded content through the FN is larger than that through the VC when $1/\lambda_{d}$ increases. This means that the denominator rapidly increases than the numerator in Equation (Equation 8) according to the increasing $1/\lambda_{d}$. In addition, at the same $1/\lambda_{d}$, the differences in the offloading efficiency between the two schemes increase as $1/\mu_{v}$ increases. This is because the effect of cases (b) and (c) in Figure 2 on the offloading efficiency is marginal at smaller $1/\mu_{v}$ and becomes significant at bigger $1/\mu_{v}$. On the other hand, from Figure 3b, it can be found that the effective VC offloading time becomes higher when $1/\lambda_{d}$ increases for both systems because the opportunity to offload the service to VC also increases. However, the increasing rate of the non-migration system is much smaller than that of CSV. This means that the cases (d) and (e) in Figure 2 do not have an important effect on the expected VC offloading time according to the change of $1/\lambda_{d}$. Specifically, Table 2 summarizes the ratio of the partial offloading time for each case (i.e., $E_{b,C}$, $E_{c,C}$, $E_{d,C}$, and $E_{e,C}$ as defined in Section 3) to the total expected offloading time (i.e., $E[t_{off}]$) for CSV. From Table 2, it can be noted that as $1/\lambda_{d}$ increases, the effect of $E_{d,C}$ and $E_{e,C}$ decreases while that of $E_{a,C}$ and $E_{b,C}$ increases. For example, when $1/\mu_{v}=0.6/\mu_{f}$, the ratio of $E_{a,C}$ to the total expected offloading time accounts for 11.7% and 21.7% for $1/\lambda_{d}=0.1$ and $1/\lambda_{d}=0.2$, respectively (i.e., approximately 10% increase). On the other hand, the ratio of $E_{d,C}$ to the total expected offloading time becomes 26.5% from 35.3% when $1/\lambda_{d}$ changes into 0.2 from 0.1 (i.e., approximately 8.8% decrease).

Table 3 shows the effect of the service migration time $t_{m}$ to the VC offloading efficiency $E_{E}$ when $1/\mu_{v}$ changes from $0.5/\mu_{f}$ to $1/\mu_{f}$. From Table 3, it can be found that the VC offloading efficiency decreases according to the increasing migration time because the service cannot be provided during the migration time. In addition, Table 4 shows the VC offloading failure probability according to the service migration time. The VC offloading failure probability increases as the migration time increases, which can be considered a risk to try to offload the service that exists. However, as the service migration has become efficient and fast [12], it can be expected that the effect of the migration time will become trivial. In addition, since the service operator can predict the migration time based on the network status [12], service migration can be exploited at low offloading failure probability.

Figure 4 illustrates the effect of the popularity rank $r_{q}$ of the content on the VC offloading efficiency $E_{E}$. We assume that the first rank is the lowest rank. For both systems, the VC offloading efficiency decreases as the popularity rank increases. This is because the probability that the VC and user have the same content decreases when the popularity rank increases. In addition, the VC offloading efficiency increases according to the increasing exponent of the Zipf distribution (i.e., ϵ). The reason is that the request probability of high-rank contents becomes higher as the exponent increases. Specifically, the sum of the request probabilities of the top ten contents becomes approximately 13% higher from ϵ=0.7 to ϵ=1.

Table 5 shows the effect of the total number of contents *Q* to the VC offloading efficiency $E_{E}$. For both systems, the VC offloading efficiency decreases as the total number of contents increases. The reason is that the probability that VC has the same content with user’s request becomes reduced when the total number of contents increases. Furthermore, the difference of the VC offloading efficiency between both schemes decreases according to the increasing the total number of contents. This is because the decreasing rate of the Zipf distribution becomes reduced according to the total number of contents.

Figure 5 illustrates the effect of the caching capacity of VC $C_{v}$ on the VC offloading efficiency $E_{E}$. For both systems, the VC offloading efficiency increases as the caching capacity of VC increases. This is because the probability that the VC has the same content with user’s request increases when the caching capacity increases. In addition, the increasing rate of CSV is higher than that of the non-migration scheme because CSV has more opportunity to utilize VC compared to the non-migration scheme. Specifically, the increasing rates of CSV and non-migration schemes are about 0.016 and 0.005, respectively, when $1/\lambda_{d}$ is 0.1 and $C_{v}$ changes from 5 to 14.

## 5. Conclusions

In this paper, we introduced a CSV where the content service is offloaded to the VC to reduce the load of the FN and provide service continuity. To analyze the performance of CSV, we developed the analytical model of the VC offloading efficiency. Simulation results show that CSV can improve the VC offloading efficiency compared to the non-migration system and the offloading efficiency is affected by the session duration, VC contact time, migration time, and the popularity rank of the content. In our future work, we will validate the CSV by means of the emulation-based feasibility test.

## Figures and Tables

**Figure 1 sensors-22-07967-f001:**
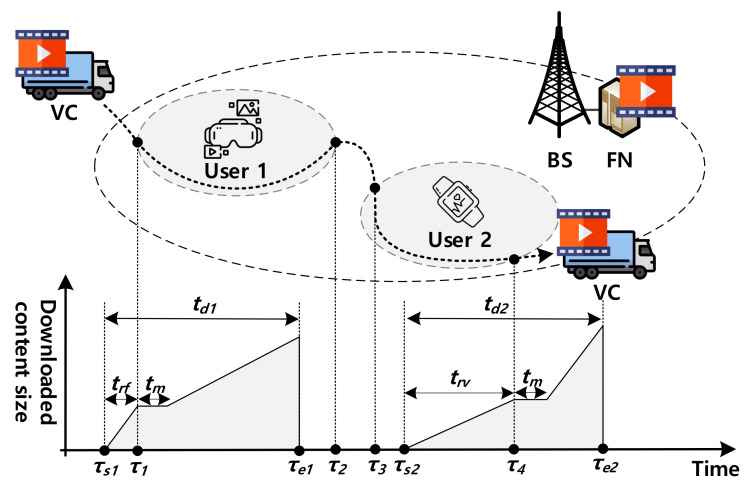
System model.

**Figure 2 sensors-22-07967-f002:**
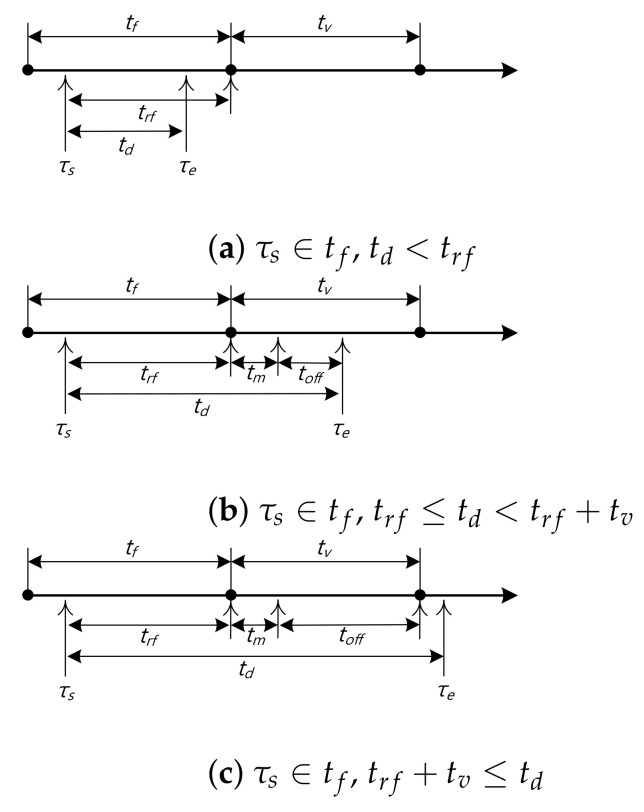
Timing diagrams of CSV.

**Figure 3 sensors-22-07967-f003:**
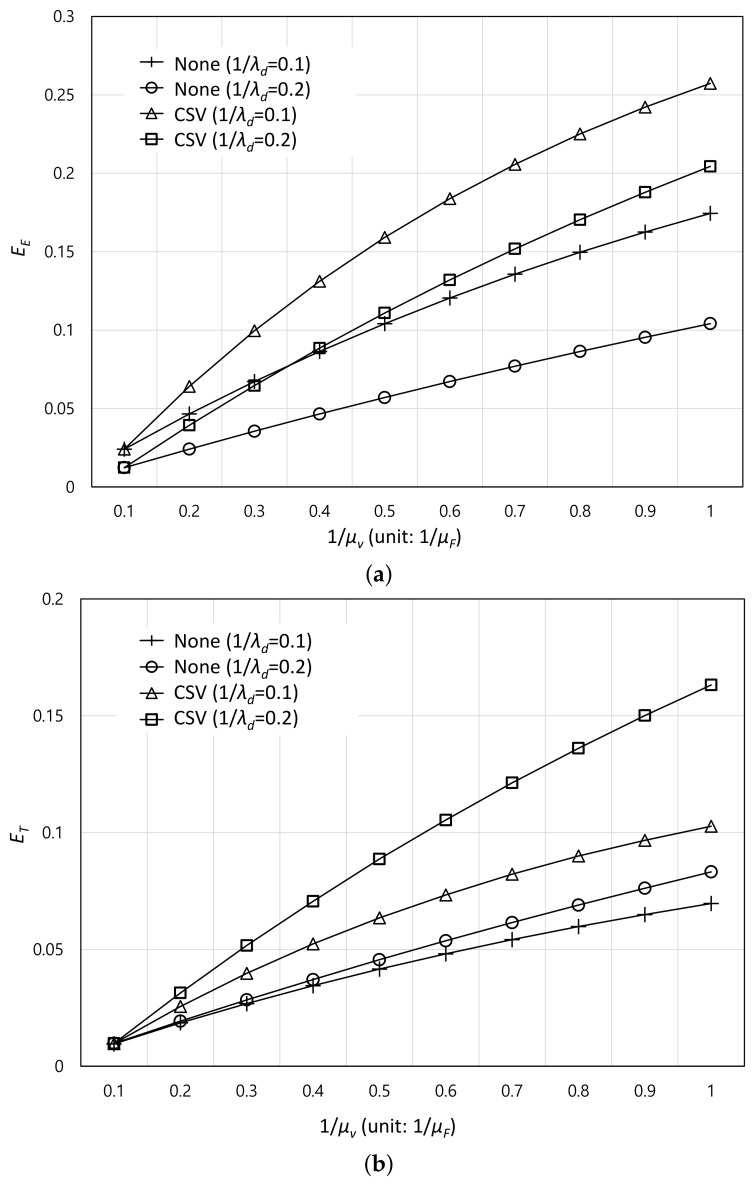
Effect of the average contact time between VC and the user. (**a**): VC Offloading Efficiency, (**b**): Effective VC Offloading Time.

**Figure 4 sensors-22-07967-f004:**
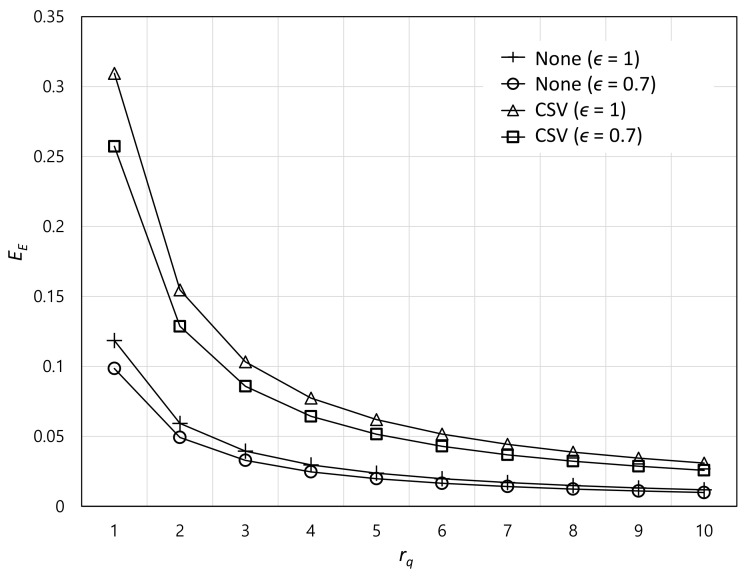
Effect of the popularity rank of the content on the VC offloading efficiency.

**Figure 5 sensors-22-07967-f005:**
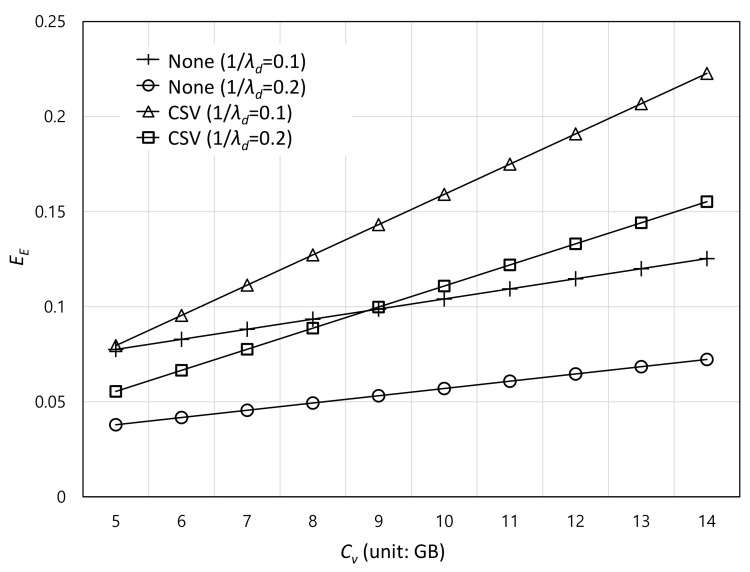
Effect of the caching capacity of VC ($C_{v}$) on the VC offloading efficiency ($E_{E}$).

**Table 1 sensors-22-07967-t001:** Simulation parameters.

Parameter	Default Value
$\mu_{f}$	10
$\lambda_{d}$	10
$d_{b}$	10 Mbps
$d_{v}$	5 Mbps
*Q*	50
$t_{m}$	0.01
$r_{q}$	1
$C_{v}$	10 GB
Content size	[500 MB, 1000 MB]
ϵ	0.7

**Table 2 sensors-22-07967-t002:** Ratio of the partial offloading time.

$1/\lambda_{d}$	$1/\mu_{v}$	$E_{b,C}$	$E_{c,C}$	$E_{d,C}$	$E_{e,C}$
0.1	$0.3/\mu_{f}$	10.1%	22.6%	34.9%	32.4%
	$0.6/\mu_{f}$	11.7%	22.6%	35.3%	30.3%
	$0.9/\mu_{f}$	12.5%	20.4%	37.4%	29.7%
0.2	$0.3/\mu_{f}$	19.1%	25.9%	28.0%	26.9%
	$0.6/\mu_{f}$	21.7%	27.3%	26.5%	24.5%
	$0.9/\mu_{f}$	22.8%	26.4%	26.8%	23.9%

**Table 3 sensors-22-07967-t003:** VC offloading efficiency ($E_{E}$) according to $t_{m}$.

	*t_m_*	0.01	0.02	0.03	0.04	0.05
1/*μ_v_*						
0.5/*μ_f_*		0.159	0.141	0.127	0.114	0.104
1/*μ_f_*		0.257	0.240	0.225	0.213	0.202

**Table 4 sensors-22-07967-t004:** VC offloading failure probability ($P_{F}$) according to $t_{m}$.

	*t_m_*	0.01	0.02	0.03	0.04	0.05
1/*μ_v_*						
0.5/*μ_f_*		0.33%	1.64%	2.93%	4.21%	5.46%
1/*μ_f_*		0.05%	0.64%	1.45%	2.25%	3.05%

**Table 5 sensors-22-07967-t005:** VC offloading efficiency ($E_{E}$) according to the total number of contents (*Q*).

	*Q*	10	30	50	70	90

CSV		0.244	0.179	0.159	0.148	0.141
None		0.160	0.117	0.104	0.097	0.092

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
