# Peer review of "CSV: Content Service Offloading System with Vehicular Caching"

_sensors, 2022, doi:10.3390/s22207967_

Round 1
Reviewer 1 Report
The paper “Performance Analysis of Content Service Offloading System with Vehicular Caching” presents an analytical model for offloading efficiency using a vehicular caching approach. The paper is very concise and the analyze is based on some simulation. The model description contains some useful details. However, there are some flows that should be addressed by the authors.
1. [44] “To offload more content services to VC with service continuity, we introduce the content service offloading system with VC, named CSV.” It not clear if the intention of the authors was to propose the CSV in this paper or it exists in some previous reference. Therefore, the authors should describe the model here and modify the title accordingly, or they should cite the references.
2. [60] “the contents that users request can exist or not in VC according to the popularity of contents and the limited caching capacity of VC”. The model and the simulation presented here does not consider the available space on the VC side, which can be on a huge importance considering VC capacity versus possible content. Indeed, the authors mentioned [100] “Moreover, due to the limited caching capacity of VC (i.e., Cv), the least-recently-used (LRU) is assumed as it is generally utilized in the content caching process for the replacement policy”. However, the capacity of VC versus the size of Q or the interaction between LRU mechanism and downloading intervals are not considered in the proposed model, which, in my opinion can make the model irrelevant in real world applications.
3. The simulation is very briefly described. No details are provided on the simulation environment, and few details on the simulation parameters. The analyze does not consider any content distribution strategies (other than CSV). However, these strategies can influence hardly the efficiency of using VC.
4. It is not clear the relation between the authors’ previous paper:
- Kyung, Yeunwoong, Eunchan Kim, and Taewon Song. "Opportunistic offloading scheme for content delivery service using electro‐mobility networks." IET Intelligent Transport Systems (July 2022),
and the proposed work. Considering the abstract of the mentioned paper “Vehicular caching (VC) in electro-mobility networks has become promising for supporting the needs of low end-to-end service delays and reducing the load of networks (i.e. edge caching [EC]) for content delivery service. However, since VCs are available only when they are in the vicinity of the user, intermittent connectivity should be considered. In this paper, the opportunistic offloading scheme for content delivery service is proposed, and a decision is made regarding whether to use VC or EC for the content delivery service. In the proposed scheme, when VC is not available, the user determines whether to choose VC or EC for the content download. …”, I consider that authors should cite this reference and explain the relation.
Author Response
We really appreciate your time and valuable comments. We have carefully revised our
paper taking into consideration of your comments and suggestions. The following attachment is
detailed responses to your comments (C: Comment, A: Answer). Please note that
modifications are marked in “blue” in the revised manuscript.

Reviewer 2 Report
Comment 1: The paper analyzes the performance of service offloading using vehicular caching. The paper is well written.
Comment 2: motivation is described as the analysis performance of the VC offloading. What does the paper want to improve? What exactly the novelty of the paper: what is being proposed? What is the network scenario for CSV, is it 5G or 6G?
In the abstract, the authors aim to introduce a CSV but there is no explanation about CSV in the paper. The contribution in Section 1-Introduction focuses on (1) analytical study and (2) simulation. In section 2, it should be explained about the CSV, is it the paper’s novelty?
Comment : how the user predict/estimate the VC position?
Author Response

(The authors gave the same response as above.)

Reviewer 3 Report
This paper brings an interesting analysis, which could enrich the state-of-the-art in the studied field. The level of language and presentation is good enough. The paper is of appropriate length, the scientific content is well organized. Based on mentioned advances I recommend to accept and publish the manuscript as it is. Nice work.
Author Response
We really appreciate your time and valuable comments.
Round 2
Reviewer 1 Report
The revised version of the paper “CSV: Content Service Offloading System with Vehicular Caching” (formerly “Performance Analysis of Content Service Offloading System with Vehicular Caching”) clarifies some aspects related to the purpose of the presented research. The new tile reflects better the content, and some missing details were added to descriptions of the method and simulation. The authors succeeded to also clarify the relation with some previous work (ref. [Kyung22]). The overall improved quality of the manuscript allows me to recommend it for publication.
Reviewer 2 Report
- The authors addressed all the points I raised and I have no further questions.